# PINK: UNVEILING THE POWER OF REFERENTIAL COMPREHENSION FOR MULTI-MODAL LLMS

## ABSTRACT

Multi-modal Large Language Models (MLLMs) have shown remarkable capabilities in many vision-language tasks. Nevertheless, most MLLMs still lack the Referential Comprehension (RC) ability to identify a specific object or area in images, limiting their application in fine-grained perception tasks. This paper proposes a novel method to enhance the RC capability for MLLMs. Our model represents the referring object in the image using the coordinates of its bounding box and converts the coordinates into texts in a specific format. This allows the model to treat the coordinates as natural language. Moreover, we construct the instruction tuning dataset with various designed RC tasks at a low cost by unleashing the potential of annotations in existing datasets. To further boost the RC ability of the model, we propose a self-consistent bootstrapping method that extends dense object annotations of a dataset into high-quality referring-expression-bounding-box pairs. The model is trained end-to-end with a parameter-efficient tuning framework that allows both modalities to benefit from multi-modal instruction tuning. This framework requires fewer trainable parameters and less training data. Experimental results on conventional vision-language and RC tasks demonstrate the superior performance of our method. For instance, our model exhibits a 12.0% absolute accuracy improvement over Instruct-BLIP on VSR and surpasses Kosmos-2 by 24.7% on RefCOCO_val under zero-shot settings. We also attain the top position on the leaderboard of MMBench. We will release the models, datasets, and codes for further research.

## 1 INTRODUCTION

Large Language Models (LLMs) (Brown et al., 2020; Raffel et al., 2020; Touvron et al., 2023a; Scao et al., 2022) show impressive capabilities across a wide range of natural language tasks. These inspiring results by LLMs have motivated researchers to extend LLMs to Multi-modal Large Language Models (MLLMs) by integrating LLMs with additional modalities, *e.g.*, image, audio, or point cloud. Visual instruction tuning (Liu et al., 2023b; Dai et al., 2023; Ye et al., 2023), using high-quality image-text instruction tuning data, allows the incorporation of visual comprehension ability into LLMs by projecting visual features into the natural language space of the LLMs. Powered by those methods, existing MLLMs are capable of basic image-level comprehension. However, they are still confronted by Referential Comprehension (RC), namely in identifying a specific object or area within an image, as illustrated in Fig. 1 (b). Understanding the reference to a particular object using the coordinates of a bounding box remains a challenge to MLLMs. This limitation significantly restricts the potential applications of MLLMs in fine-grained understanding of images.

This paper introduces a novel MLLM named **Pink** 🔲[1] with an enhanced RC ability. To enable the LLM to understand bounding box coordinates, we convert the coordinates into texts in a specific format, allowing the LLM to treat them as natural language and use them as input and output as shown in Fig. 1. The model comprises a visual encoder, a projection layer to bridge the two modalities, and an LLM. The visual encoder is responsible for image representations. Its capability directly determines the performance of the MLLM. As discussed in Wang et al. (2023), tuning the visual encoder with a small-scale visual instruction tuning dataset may lead to semantic loss, which is harmful to the image representation capability of the visual encoder. Consequently, some visual

---

[1]This name is from the main character of the album *The Wall* by the great rock band *Pink Floyd*.

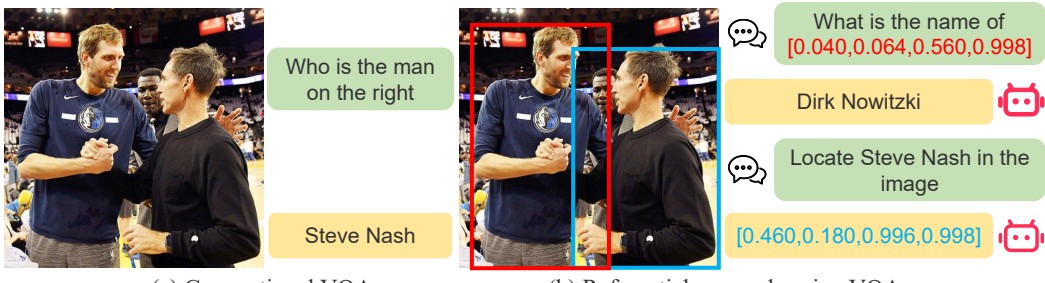

(a) Conventional VQA  (b) Referential comprehension VQA

Figure 1: Comparison between the conventional VQA and referential comprehension VQA. In referential comprehension VQA, a model is required to understand questions about a specific area in the image provided by the coordinates of a bounding box and can also mention a particular area in the response if needed. Examples shown in (b) are generated by our model.

instruction tuning methods (Zhu et al., 2023; Liu et al., 2023b) freeze the visual encoder and only fine-tune the projection layer. This design does not require large-scale image-text pairs but may result in sub-optimal alignment between two modalities. Other methods (Ye et al., 2023; Dai et al., 2023) fine-tune the visual encoder to better align two modalities. These methods require millions or even billions of image-text pairs. Furthermore, these methods still need to freeze the visual encoder during instruction tuning to prevent its semantic loss. To enhance both visual and language modalities by multi-modal instruction tuning, we leverage a parameter-efficient tuning framework, which inserts several Adapters (Houlsby et al., 2019) into both visual encoder and LLM. We hence fine-tune the Adapters and the projection layer during visual instruction tuning, meanwhile freezing the main parameters of the model to avoid forgetting the learned knowledge. Adapters provide a way to adapt the model to the multi-modal instruction tuning.

The instruction tuning dataset plays a crucial role in training MLLMs. Existing datasets only offer limited RC tasks like visual grounding, grounding caption (Kazemzadeh et al., 2014), and pointQA (Mani et al., 2020). Solely relying on these tasks limits the instruction following ability of the model on RC tasks. To increase the diversity of RC tasks, we propose a new dataset construction pipeline that extends the annotations of existing datasets to various RC tasks. Specifically, we design several RC tasks, such as visual relation reasoning and visual spatial reasoning, based on the annotations of existing datasets like Visual Genome (Krishna et al., 2017). To further involve more training data about RC tasks, we introduce a self-consistent bootstrapping method that extends dense object annotations to referring-expression-bounding-box pairs. Different from other methods (Chen et al., 2023b; Zhao et al., 2023) that rely on the API of GPT4 (OpenAI, 2023) to generate instruction tuning data for RC tasks, our model achieves impressive capability in both image-level comprehension and referential comprehension using available annotations from existing datasets.

Extensive experiments on conventional vision-language and RC tasks demonstrate the superior performance of our method. For instance, with only 6.7M tunable parameters, we achieve up to 6.0% absolute accuracy improvement on OK-VQA (Marino et al., 2019) compared to Shikra (Chen et al., 2023b). We also attain the top position on the leaderboard of MMBench (Liu et al., 2023c), even surpassing methods that rely on more training data and advanced LLMs.

Our work is an original exploration to enhance LLMs with RC abilities by leveraging annotations from existing datasets and alleviating the dependence on expensive GPT4 APIs. The proposed instruction tuning dataset construction method exposes the potential of annotations from existing datasets and increases the diversity of RC tasks without reliance on the API of GPT4. Our self-consistent bootstrapping method provides a new solution for extending existing visual datasets into multi-modal datasets. The tuning framework allows that both modalities benefit from multi-modal instruction tuning while relaxing the dependence on large-scale image-text pairs. It also reduces the number of tunable parameters to 6.7M. We will release the codes and the generated datasets to facilitate further research and evaluation.

## 2 RELATED WORKS

**Multi-modal Large Language Model.** Several approaches have been proposed to condition LLMs with additional modalities. Flamingo (Alayrac et al., 2022) proposes Perceiver to extract representative visual tokens and leverages cross-attention to condition LLMs. Q-Former is proposed in BLIP-2 (Li et al., 2023b) to align visual features with LLMs. Building upon the idea of instruction tuning (Wei et al., 2021), some works perform multi-modal instruction tuning to improve the ability of MLLMs to follow instructions. Mini-GPT4 (Zhu et al., 2023) constructs a high-quality instruction tuning dataset and fine-tunes only a single fully connection layer. LLaVA (Liu et al., 2023b) prompts GPT4 (OpenAI, 2023) to generate visual instruction data and aligns the visual and language modalities with a single projection layer. LLaMA-Adapter V2 (Gao et al., 2023) introduces a new adapter module and inserts it into the LLM. During training, the adapter modules, normalization layers, layer biases and layer scales in the LLM are updated. Freezing the visual encoder during two modalities alignment training stage reduces requirement of these methods to large-scale image-text pairs. Other methods leverage millions of image-text pairs to achieve better alignment between the two modalities. Instruct-BLIP (Dai et al., 2023) introduces an instruction-aware visual feature extraction method and fine-tunes the entire Q-Former, showing great zero-shot generalization ability on various multi-modal tasks. mPlug-Owl (Ye et al., 2023) incorporates a visual abstractor module to align the two-modalities. Both the visual encoder and the visual abstractor are updated during the pre-training stage. For instruction tuning, low-rank adaption (Hu et al., 2022) is integrated to fine-tune the LLMs. All methods freeze the visual encoder during multi-modal instruction tuning, making it cannot benefit from the multi-modal instruction tuning.

**Referential Comprehension of MLLMs.** In daily human communication, it is common to refer to objects or regions in a scene. Therefore, enhancing MLLMs with the RC ability is highly valuable. Inspired by Pix2Seq (Chen et al., 2021), many works use discrete coordinate tokens to encode spatial information and unify RC tasks as sequence generation tasks, *e.g.*, OFA (Wang et al., 2022), Unified-io (Lu et al., 2022), and Kosmos-2 (Peng et al., 2023). Another line of works, as seen in PVIT (Chen et al., 2023a) and GPT4RoI (Zhang et al., 2023), leverage the ROI operation (He et al., 2017) to extract features of referring objects. These works require extra modules and may lose context information because of the ROI operation. More importantly, these works cannot give answers with referring objects, limiting their applications, *e.g.*, visual grounding. In addition to the model design, the construction of RC instruction tuning data also plays a crucial role. Shikra (Chen et al., 2023b) converts existing datasets of RC tasks, *e.g.*, visual grounding (Kazemzadeh et al., 2014) and pointQA (Mani et al., 2020), into the instruction following format. Kosmos-2 uses the grounding model GLIP (Li et al., 2022) to extract coordinates of noun chunks in image captions and constructs a large-scale datasets. Similar to LLaVA, ChatSpot (Zhao et al., 2023), PVIT (Chen et al., 2023a), and Shikra (Chen et al., 2023b) all prompt GPT4 to generate instruction tuning data for RC.

Existing methods for enhancing MLLMs with the RC ability typically incorporate existing RC datasets or rely on the API of GPT4. Most datasets only contain a limited number of RC tasks, while the use of GPT4 API is expensive and uncontrollable. Differently, our work explores the potential of existing datasets, *e.g.*, Visual Genome (Krishna et al., 2017), by designing more RC tasks using their annotations. Furthermore, we propose a self-consistent bootstrapping method to extend the dense object annotations in the dataset to referring-expression-bounding-box pairs. Our improved dataset construction method leads to superior performance in both zero-shot and fine-tuning evaluation settings.

## 3 METHODOLOGY

### 3.1 MODEL ARCHITECTURE AND TRAINING PIPELINE

**Model Architecture.** As shown in Fig. 2, Pink consists of a visual encoder $\Phi_V$, a projection layer $\Phi_P$, and a decoder-only LLM $\Phi_L$. Given an image $I$ and a sequence of word embeddings $Q_T$ representing an instruction sentence, the visual encoder is employed to embed the image as a sequence of visual tokens $Z_V = \Phi_V(I)$. A linear layer is used as $\Phi_P$ to convert $Z_V$ into the input space of the LLM $Z_T = \Phi_P(Z_V)$. $Z_T$ and $Q_T$ are concatenated and fed into $\Phi_L$ to generate the next word.

To enable the LLMs to take coordinates as input and output, similar to Shikra (Chen et al., 2023b), we convert coordinates into texts in a specific format. Specifically, for a bounding box represented

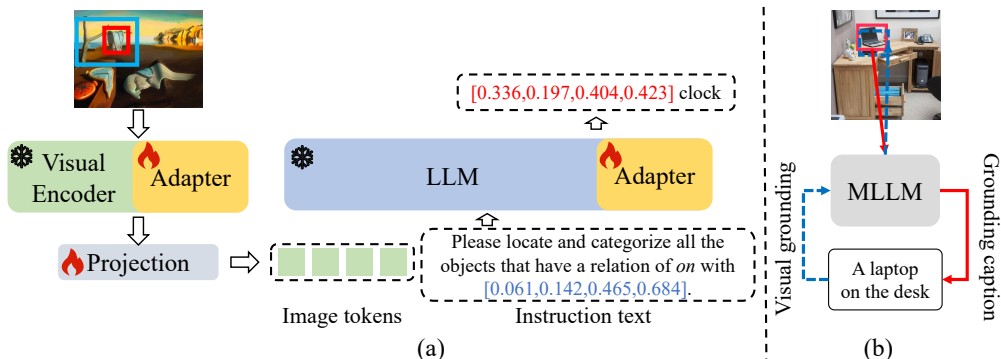

Figure 2: The illustration of (a) our model and (b) self-consistent bootstrapping method. Our model consists of three main components: a visual encoder, a projection layer, and a decoder-only LLM. The coordinates of a bounding box are converted into texts in a specific format. During instruction tuning, we freeze the visual encoder and LLM and only update the Adapters and the projection layer, making both the visual and language modalities benefit from the multi-modal instruction tuning. Given a bounding box, self-consistent bootstrapping method first generates its description by asking MLLM to perform grounding caption, then leverages the visual grounding to locate the generated description. The low-quality description will be filtered out if the IOU between the predicted and ground-truth bounding box is below a threshold.

by its coordinates of the top-left and bottom-right corners $[x_{min}, y_{min}, x_{max}, y_{max}]$, we normalize the coordinates to the range $[0, 1]$ with respect to the image size and retain 3 decimal places for each number, *e.g.*, [0.222,0.333,0.444,0.555]. This design allows the coordinates to be processed as regular text and can appear in both the input and output of the model.

Directly fine-tuning the entire visual encoder requires millions of image-text pairs to avoid semantic loss due to the limited instruction tuning data (Wang et al., 2023). To mitigate this issue and enable the visual encoder to benefit from multi-modal instruction tuning, we freeze the visual encoder meanwhile introducing tunable modules into it. This approach prevents the visual encoder from suffering semantic loss and provides an efficient way to adapt the model. In particular, we employ the Adapter (Houlsby et al., 2019) at both the visual encoder and LLM. Given an input token feature $Z \in \mathbb{R}^d$, the architecture of an Adapter is defined as follows,

$$\hat{Z} = \sigma\left(ZW_d\right)W_u + Z, \tag{1}$$

where $W_d \in \mathbb{R}^{d \times d_s}$ and $W_u \in \mathbb{R}^{d_s \times d}$ denote the weight matrices, $d_s$ is the hidden dimension which is much smaller than $d$, and $\sigma$ denotes the non-linear activation function. $W_u$ is initialized to zero to ensure that at the beginning of the training, the Adapter does not change the original output.

**Training Pipeline.** Our method maps both the image and coordinates into the input space of the LLM. Consequently, the model can be trained end-to-end using a language modeling task, which predicts the next word token based on the preceding context.

The model is trained in two stages. In the first stage, we exclusively fine-tune the projection layer with a small set of image-text pairs. In the second stage, we freeze both the visual encoder and LLM and fine-tune the newly added Adapters and the projection layer with the instruction tuning dataset. Therefore, both the visual and language modalities can benefit from the multi-modal instruction tuning. Compared with fine-tuning the entire model, we only updates a small number of parameters, and does not require a large-scale dataset with image-text pairs.

## 3.2 INSTRUCTION TUNING DATASET CONSTRUCTION

To create the instruction tuning dataset, we unify all the multi-modal tasks into a vision-language dialogue format:

> **Image:** {Image tokens}
> **User:** {Instruction template}
> **Assistant:** {Response}

where the placeholders {Image tokens}, {Instruction template}, and {Response} will be replaced with the image tokens extracted by $\Phi_V$, task instruction template, and the response, respectively.

It is important to introduce diverse RC tasks for the instruction tuning to enhance the instruction-following ability of MLLMs on RC tasks. Existing datasets only offer limited RC tasks, *e.g.*, visual grounding, grounding caption (Kazemzadeh et al., 2014), and pointQA (Mani et al., 2020), which are insufficient to cover a wide range of RC abilities MLLMs desire to have. Therefore, besides the RC tasks mentioned above, we design more diversified RC tasks by incorporating annotations from Visual Genome (Krishna et al., 2017), which contain information about region descriptions, objects, and relations between different objects. Following parts proceed to introduce those tasks.

**Visual Relation Reasoning.** Visual Genome has annotated millions of relationship triplets (*subject-predicate-object*), *e.g.*, man-wearing-hat. We design two types of visual relation reasoning tasks by leveraging these annotations to help the model understand the relationship between different objects: (1) We randomly select a relationship triplet. Given the coordinates of *subject* and *object*, the model is required to predict their relation. (2) We randomly select one *subject* and a relation from the annotations. The model is required to detect all objects that have selected relation with *subject* and output their coordinates and class names.

**Coarse Visual Spatial Reasoning.** We introduce a coarse visual spatial reasoning task by utilizing the object annotations from Visual Genome. This task enhances the visual spatial reasoning ability of the MLLMs. We define four coarse spatial positions as **top-left, top-right, bottom-left, and bottom-right**. Given a randomly selected object and a coarse spatial position, the model is required to identify all objects located at this position relative to the selected object and predict their coordinates and class names.

**Object Counting.** To endow the model with the concept of different instances and the capability of fine-grained object recognition, we design an object counting task. This task asks the model to count objects in the image that belong to the same category as the given object or class name.

**Object Detection.** Object detection can empower the model to locate the position and boundaries of objects. Given a class name or a selected object, the model is asked to identify all objects that belong to the same category as the given object or class name and provide their coordinates.

By incorporating these RC tasks into the instruction tuning, the model can substantially learn a variety of RC abilities. To clarify the designed tasks, we list some example instruction templates,

> **Visual Relation Reasoning:**
> **User:** Assist me in finding the relation between <subject> and <object> in the photo.
> **Assistant:** <relation>.
>
> **User:** Please locate and categorize all the objects that have a relation of <relation> with <subject>.
> **Assistant:** <object> <category> <object> <category>.
>
> **Coarse Visual Spatial Reasoning:**
> **User:** Identify the objects located at <loc> of <object>.
> **Assistant:** <object> <category> <object> <category>.
>
> **Object Counting:**
> **User:** How many objects in the image are of the same category as <object>.
> **Assistant:** <number>.
>
> **Object Detection:**
> **User:** Identify all the objects that fit the same category as <object> and display their

> coordinates.
> **Assistant:** <object> <object>.

where the placeholders <object> and <category> will be replaced with the bounding box coordinates and the class name of a referring object, respectively. <subject> will be replaced with the bounding box coordinates of the selected subject. <relation>, <loc>, and <number> will be replaced with the relation between different objects, the selected relative spatial position, and the number of the objects, respectively. All instruction templates can be found in Appendix A.5.

### 3.3 SELF-CONSISTENT BOOTSTRAPPING METHOD

The constructed instruction-following datasets are adopted to reinforce the basic RC ability of MLLMs. We also would like to acquire more high quality data at low cost to further boost the RC ability of the model. Existing datasets for object detection provide valuable bounding box annotations for objects appearing in the image, making them promising resources for instruction tuning dataset construction. To integrate these datasets into the multi-modal instruction tuning, we propose a self-consistent bootstrapping method by leveraging the model itself. This method extends the bounding box annotations to the referring-expression-bounding-box pairs. It comprises two key stages: bounding box description bootstrapping and self-consistent filtering as shown in Fig. 2 (b).

At the bounding box description bootstrapping stage, given a bounding box $B$ of an object, we prompt the model to generate a description $D_B$ for that object by leveraging its ability of grounding caption. Due to the complexity of scenes or the presence of similar duplicate objects, the generated description may be noisy or fails to uniquely describe the corresponding object. Then, the self-consistent filtering stage focuses to filter out those low-quality descriptions. Specifically, with the generated description $D_B$, we locate this description in the image and predict the bounding box $\hat{B}$ by leveraging the visual grounding ability of our model. The generated description will be removed if the Intersection Over Union (IOU) between $B$ and $\hat{B}$ is below a pre-defined threshold $\lambda$. This stage ensures that only high-quality descriptions are retained.

These two stages are performed to extend every annotated object in the dataset with description. This extended dataset is then well-suited for a wide range of RC tasks, *e.g.*, coarse visual spatial reasoning, object detection, object counting, visual grounding and grounding caption. Illustrations of generated data are in Appendix A.4. This self-consistent bootstrapping method serves as a powerful tool to harness the potential of object detection datasets for enhancing the RC ability of our model.

## 4 EXPERIMENT

### 4.1 EXPERIMENTAL SETTING

**Model Architecture.** We employ the ViT-L/14 (Dosovitskiy et al., 2021) as the visual encoder, which is pre-trained with CLIP (Radford et al., 2021). We choose an instruction-tuned model Vicuna-7B (Vicuna, 2023) as the LLM. The projection layer is a single fully connection layer. The Adapters are inserted before each self-attention layer of both the visual encoder and the LLM, with a hidden dimension $d_s = 8$. The tunable parameter numbers of Adapter in the visual encoder and LLM are 393,216 and 2,097,152, respectively. The number of parameters in the projection layer is 4,194,304. Therefore, the total number of tunable parameters is about 6.7M.

**Training Data.** The first stage utilizes 595K image-text pairs from CC3M (Sharma et al., 2018), the same as LLaVA (Liu et al., 2023b). The second stage adopts VQAv2 (Goyal et al., 2017), LLaVA-150K (Liu et al., 2023b), A-OKVQA (Schwenk et al., 2022), Flickr30K (Plummer et al., 2015), Visual Genome (Krishna et al., 2017) and Object365 (Shao et al., 2019) with referring-expression-bounding-box pairs generated by our self-consistent bootstrapping method. At each training iteration, when using an image in Visual Genome or Object365, one designed RC task will be selected randomly and converts to the instruction-following format. The model used to generate referring-expression-bounding-box pairs in Object365 is trained with the aforementioned datasets, excluding Object365 itself. Note that we reduce the probability of sampling Object365 in batch construction

Table 1: Ablation study on instruction tuning dataset construction and training settings of visual encoder under a zero-shot setting. "Baseline" denotes leveraging Visual Genome by only performing visual grounding and grounding caption tasks. "VG" denotes Visual Genome. "R", "S", "C", and "D" denote the visual relation reasoning, coarse visual spatial reasoning, object counting and object detection tasks, respectively. † denotes generated referring-expression-bounding-box pairs in Object365 are not filtered with the self-consistent method. "Freezing" and "Full-tuning" denotes freezing the visual encoder and training the entire visual encoder, respectively. "LoRA" denotes using LoRA instead of the Adapter to perform parameter-efficient tuning.

| Settings | IconQA | VSR | OK-VQA | RefCOCO_val | Local | LookTwice |
|---|---|---|---|---|---|---|
| Baseline | 44.6 | 65.6 | 58.5 | 55.0 | 0.0 | 0.2 |
| w/o VG | 43.1 | 62.8 | 58.3 | - | - | - |
| + R | 44.4 | 65.7 | 58.5 | 52.1 | 17.1 | 12.8 |
| + R,S | 46.2 | 65.8 | 58.5 | 52.7 | 50.9 | 60.0 |
| + R,S,C | 47.4 | 65.7 | 58.9 | 53.1 | 53.4 | 60.7 |
| + R,S,C,D | **47.8** | 66.3 | 59.5 | 54.1 | 54.6 | 63.1 |
| + R,S,C,D + Object365† | 44.6 | 65.9 | 58.7 | 73.8 | 52.1 | 69.2 |
| + R,S,C,D + Object365 | 47.7 | **67.1** | **59.5** | **77.0** | **57.2** | **70.3** |
| Freezing | 42.9 | 61.5 | 58.3 | 37.2 | 44.9 | 57.5 |
| Full-tuning | 36.9 | 48.6 | 33.1 | 0.05 | 26.1 | 54.1 |
| LoRA | 44.3 | 65.4 | 58.9 | **54.7** | **56.7** | 62.2 |
| Our | **47.8** | **66.3** | **59.5** | 54.1 | 54.6 | **63.1** |

to avoid a large number of training samples in Object365 dominating the training. The summary of the datasets can be found in Appendix A.2.

**Training Details.** AdamW is adopted as the optimizer. In the first stage, the model is trained for 1 epoch with a batch size of 128 and weight decay of 0.0. After a warm-up period of 200 steps, the learning rate starts at 0.03 and decays to 0 with the cosine schedule. In the second stage, the model is trained for 6 epochs with a batch size of 32 and weight decay of 0.05. The warm-up phase consists of 10k steps and the learning rate starts at 5e-4. The input image is resized to $224 \times 224$ without any additional data-augmentation. We set $\lambda$ as 0.5 to filter out low-quality descriptions. The model is trained using 8 NVIDIA A100 GPUs.

**Evaluation Settings.** We evaluate our model on various datasets under the zero-shot and fine-tuning settings to validate the instruction-following ability of the trained model. These datasets encompass conventional multi-modal reasoning tasks, including abstract diagram understanding (IconQA (Lu et al., 2021)), visual spatial reasoning (VSR (Liu et al., 2023a)), knowledge-intensive VQA (OK-VQA (Marino et al., 2019)), scene understanding (GQA (Hudson & Manning, 2019)), and RC tasks such as RefCOCO/+ (Kazemzadeh et al., 2014), RefCOCOg (Mao et al., 2016), Visual-7W (Zhu et al., 2016), PointQA-Local/LookTwice (Mani et al., 2020).

## 4.2 ABLATION STUDY

**Instruction Tuning Dataset Construction.** To investigate the impact of instruction tuning dataset construction, we conduct ablation studies by excluding Visual Genome, designed RC tasks, and Object365 with referring-expression-bounding-box pairs. The results are summarized in Table 1. The RC tasks can benefit the conventional multi-modal reasoning tasks, *e.g.*, adding visual grounding and grounding caption tasks leads to an improvement of 2.8% on VSR. When the model is trained solely with visual grounding and grounding caption, it fails to provide correct responses to the questions in PointQA-Local/LookTwice, indicating limited instruction-following ability for RC tasks. As more RC tasks are included, the model begins to exhibit better instruction-following ability for these tasks. The combination of all designed RC tasks yields the best performance on both conventional multi-modal reasoning tasks and RC tasks, thus validating the effectiveness of our instruction tuning dataset construction method. Furthermore, incorporating Object365 further enhances the performance of our method on RC tasks. For example, on RefCOCO_val, the zero-shot accuracy increases from 54.1% to 77.0%. Notably, the exclusion of the self-consistent method results in the degradation of performance from 77.0% to 73.8% on RefCOCO_val due to low-quality referring-

Table 2: Comparison with other methods. "#Data" and "#Trainable Param." indicate the number of training data and trainable parameters, respectively. * denotes Object365 with generated referring-expression-bounding-box pairs is used during training.

| Models | #Data | #Trainable Param. | IconQA | VSR | OK-VQA | GQA |
|---|---|---|---|---|---|---|
| Flamingo-9B | 1B | 1.8B | - | - | 44.7 | - |
| BLIP-2 | 129M | 188M | 39.7 | 50.0 | - | 41.3 |
| Instruct-BLIP | 15.5M | 188M | 43.1 | 54.3 | - | 49.2 |
| Shikra-7B | 2.1M | 7B | 24.3 | 63.3 | 53.5 | 47.4 |
| Pink | 1.5M | 6.7M | **47.8** | **66.3** | **59.5** | **52.6** |

(a) Zero-shot results on conventional multi-modal reasoning tasks.

| Models | RefCOCO | | | RefCOCO+ | | | RefCOCOg | |
|---|---|---|---|---|---|---|---|---|
| | val | testA | testB | val | testA | testB | val | test |
| Kosmos-2 (Peng et al., 2023) | 52.3 | 57.4 | 47.3 | 45.5 | 50.7 | 42.2 | 60.6 | 61.7 |
| GRILL (Jin et al., 2023) | - | - | - | - | - | - | - | 47.5 |
| Pink | 54.1 | 61.2 | 44.2 | 43.9 | 50.7 | 35.0 | 59.1 | 60.1 |
| Pink* | **77.0** | **82.4** | **68.2** | **65.6** | **75.2** | **53.4** | **72.4** | **74.0** |

(b) Zero-shot results on visual grounding task.

| Models | RefCOCO | | | RefCOCO+ | | | RefCOCOg | | Visual-7W | LookTwice |
|---|---|---|---|---|---|---|---|---|---|---|
| | val | testA | testB | val | testA | testB | val | test | | |
| OFA-L | 80.0 | 83.7 | 76.4 | 68.3 | 76.0 | 61.8 | 67.6 | 67.6 | - | - |
| Shikra-7B | 87.0 | 90.6 | 80.2 | 81.6 | 87.4 | 72.1 | 82.3 | 82.2 | 84.3 | 72.1 |
| Pink | 88.3 | 91.7 | 84.0 | 81.4 | 87.5 | 73.7 | 83.7 | 83.7 | 85.1 | 73.5 |
| Pink* | **88.7** | **92.1** | **84.0** | **81.8** | **88.2** | **73.9** | **83.9** | **84.3** | **85.3** | **73.6** |

(c) Fine-tuning results on RC tasks.

expression-bounding-box pairs generated by the model. These results demonstrate the value of the generated data and underscore the importance of the proposed self-consistent method.

**Training Settings of Visual Encoder.** To further assess the effectiveness of our training setting for the visual encoder, we conduct experiments with different training settings in Table 1. Full-tuning the visual encoder results in a significant performance degradation. The performance on RefCOCO_val drops from 54.1% to 0.05%. This result aligns with the conclusion in Wang et al. (2023) that fine-tuning the visual encoder using a small-scale instruction tuning dataset can lead to a subsequent drop in performance. Freezing the visual encoder also leads to performance degradation on various datasets. It can be attributed to that the visual modality cannot benefit from the multi-modal instruction tuning. Our design allows for the optimization of both modalities and leverages the benefits of multi-modal instruction tuning, resulting in improved overall performance. Moreover, using LoRA (Hu et al., 2022) instead of the Adapter to perform parameter-efficient tuning can also achieve improved performance compared with Full-tuning or Freezing, demonstrating the effectiveness of adapting the visual encoder during multi-modal instruction tuning.

## 4.3 COMPARISON WITH OTHER METHODS

This section proceeds to validate the effectiveness of our method through comparison with other methods. More qualitative results can be found in Appendix A.4.

**Zero-shot Evaluation on Conventional Multi-modal Reasoning Tasks.** To evaluate the instruction-following ability of our method, we conduct experiments on several public benchmarks. These benchmarks assess various aspects of multi-modal comprehension ability of the model. As shown in Table 2a, our model achieves the best performance among the compared MLLMs with only 6.7M trainable parameters. This demonstrates that enhancing the model's fine-grained perception ability to the image via RC tasks can also benefit the conventional multi-modal reasoning tasks.

**Zero-shot Evaluation on Visual Grounding Task.** Visual grounding is a fundamental RC task that requires the model to predict the coordinates of a bounding box based on a given description. We evaluate our model on three well-established datasets under the zero-shot setting in Table 2b.

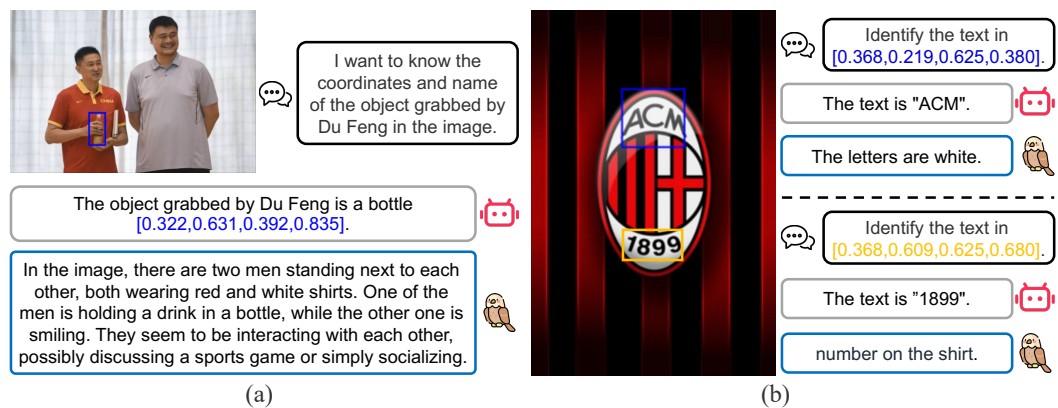

Figure 3: A comparison between Pink 🤖 and Shikra 🦉 on referential comprehension reasoning.

Our model significantly outperforms Kosmos-2, which is trained with a generated dataset GRIT containing 91M images and 115M referring expressions. This result validates that the referring-expression-bounding-box pairs generated by our proposed self-consistent bootstrapping method can effectively enhance the performance of the model on the visual grounding task.

**Comparison with Other Models under Fine-tuning Setting.** To further validate the RC ability of our method, we compare our model with other models (Wang et al., 2022; Chen et al., 2023b) that can deal with RC tasks. The comparison incorporates models that have the capability to handle various vision-language tasks. Models specifically designed for the visual grounding task are not included. Compared models are trained using the training sets of these datasets. For a fair comparison, in addition to the original training data, we also include the training sets of these datasets in the second stage of training. The results, summarized in Table 2c, show that our model obtains promising performance under fine-tuning setting. This can be attributed to the diversity of RC tasks in the instruction tuning. Moreover, Object365 can further improve the performance, even when the training sets of these tasks are already included. This demonstrates the effectiveness of our self-consistent bootstrapping method to incorporate more training data for enhancing the RC performance.

**Qualitative Results on Referential Comprehension Reasoning.** As shown in Fig. 3 (a), we design a question that requires the advanced reasoning ability of the model. To answer the question accurately, the model needs to first identify Du Feng (A famous Chinese basketball player), and then understand the action of grabbing. Our model successfully provides the correct answer. The response of Shikra is completely unrelated to the question. As shown in Fig. 3 (b), our model accurately recognizes the optical characters located at different positions in the image by providing different coordinates. Shikra also shows limited instruction-following ability. We tried many different inputs such as "What is written in [0.368,0.219,0.625,0.380].". Shikra still cannot give correct responses. Experimental results validate that Shikra has poor instruction-following ability in RC tasks and can only handle the RC tasks included in the training stage. In contrast, benefiting from various RC tasks during the instruction tuning stage, our model achieves strong instruction-following ability.

## 5 CONCLUSION

In this paper, we introduce Pink, a novel MLLM with advanced RC abilities. To enable both the visual encoder and LLM to benefit from multi-modal instruction tuning, we freeze both the visual encoder and LLM, and insert tunable Adapters into both of them. This design leads to both visual and language modalities enhanced by multi-modal instruction tuning and reduces the number of required image-text pairs. We propose a novel instruction tuning dataset construction method, which can convert annotations from existing datasets into diverse RC tasks. To acquire more instruction tuning data at low cost, we propose a self-consistent bootstrapping method that extends object annotations to the referring-expression-bounding-box pairs. Extensive experiments on various benchmarks demonstrate the superior performance of Pink on both conventional multi-modal reasoning tasks and RC tasks.

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

Table 3: Statistics of training datasets.

| Training Stage | Dataset | Data Number |
|---|---|---|
| Stage 1 | LLaVA-CC3M-Pretrain-595K (Liu et al., 2023b) | 595K |
| Stage 2 | LLaVA-150K (Liu et al., 2023b) | 150K |
| Stage 2 | VQAv2 (Goyal et al., 2017) | 444K |
| Stage 2 | A-OKVQA (Schwenk et al., 2022) | 17,056 |
| Stage 2 | Visual Genome (Krishna et al., 2017) | 108K |
| Stage 2 | Flickr30K (Plummer et al., 2015) | 150K |
| Stage 2 | Object365* (Shao et al., 2019) | 1M |

Table 4: Summary of evaluation datasets.

| Dataset | Split | Metric |
|---|---|---|
| IconQA (Lu et al., 2021) | multi-text-choice test | Accuracy |
| VSR (Liu et al., 2023a) | zero-shot test | VQA Score |
| OK-VQA (Marino et al., 2019) | val | VQA Score |
| GQA (Hudson & Manning, 2019) | test-dev | VQA Score |
| RefCOCO (Kazemzadeh et al., 2014) | val & testA & testB | Accuracy |
| RefCOCO+ (Kazemzadeh et al., 2014) | val & testA & testB | Accuracy |
| RefCOCOg (Mao et al., 2016) | val & test | Accuracy |
| PointQA-Local (Mani et al., 2020) | test-dev | VQA Score |
| PointQA-LookTwice (Mani et al., 2020) | test | VQA Score |
| Visual-7W (Zhu et al., 2016) | which box test | Accuracy |
| MMBench (Liu et al., 2023c) | test | Accuracy |

# A  APPENDIX

## A.1  DISCUSSION OF LIMITATIONS

Our method relies on the LLM. Therefore, it has some shortcomings from the LLM, such as bias or unfair response, and hallucination. We also find that our model is not good at object detection in complex scenarios, *e.g.*, identifying multiple tiny objects in the image. It may be because the input resolution of the image is low. However, increasing the input resolution of the image is not a trivial task. Many efforts are still needed to deal with such tasks.

## A.2  STATISTICS OF TRAINING AND EVALUATION DATASETS

The statistics for the training datasets and evaluation datasets are summarized in Table 3 and Table 4, respectively. We also provide an overview of the generated referring-expression-bounding-box pairs in Object365 in Table 5. To ensure the quality of data, we apply certain filters during preprocessing. Firstly, we exclude images containing more than 15 objects. Moreover, for the purpose of bounding box description bootstrapping, we only consider objects that cover an area of more than 2,000 pixels. As a result, our dataset comprises 1,063,034 images, with a total of 4,961,822 generated referring-expression-bounding-box pairs. To further enhance the reliability of our dataset, we perform a self-consistent method that filters out 2,528,619 low-quality referring-expression-bounding-box pairs.

Table 5: Statistics of generated referring-expression-bounding-box pairs in Object365.

| Images | Referring-expressions | Avg Expression Length |
|---|---|---|
| 1,063,034 | 2,433,203 | 3.6 |

Table 6: CircularEval results on MMBench test set (Liu et al., 2023c). This result is obtained from the leaderboard of MMBench at 2023/09/01.

| Models | Overall | LR | AR | RR | FP-S | FP-C | CP |
|---|---|---|---|---|---|---|---|
| Instruct-BLIP (Dai et al., 2023) | 36.0 | 14.2 | 46.3 | 22.6 | 37.0 | 21.4 | 49.0 |
| LLaVA (Liu et al., 2023b) | 36.2 | 15.9 | 53.6 | 28.6 | 41.8 | 20.0 | 40.4 |
| LLaMA-Adapter-v2 (Gao et al., 2023) | 39.5 | 13.1 | 47.4 | 23.0 | 45.0 | 33.2 | 50.6 |
| Otter-I (Li et al., 2023a) | 48.3 | 22.2 | 63.3 | 39.4 | 46.8 | 36.4 | 60.6 |
| Kosmos-2 (Peng et al., 2023) | 58.2 | 48.6 | 59.9 | 34.7 | 65.6 | 47.9 | 70.4 |
| Shikra (Chen et al., 2023b) | 60.2 | 33.5 | 69.6 | 53.1 | 61.8 | 50.4 | 71.7 |
| JiuTian (jiu, 2023) | 67.4 | 48.9 | **78.5** | 70.9 | 67.8 | 54.4 | 72.6 |
| mPlug-Owl (Ye et al., 2023) | 68.5 | 56.8 | 77.9 | 62.0 | 72.0 | 58.4 | 72.6 |
| Pink | **74.1** | **58.5** | 78.2 | **73.2** | **77.3** | **67.2** | **78.7** |

### A.3 CIRCULAREVAL RESULTS ON MMBENCH TEST SET

It is non-trivial to evaluate MLLMs since traditional benchmarks suffer from a lack of fine-grained ability assessment and non-robust evaluation metrics. To address this issue, MMBench (Liu et al., 2023c) has been proposed as a new benchmark that can evaluate various abilities of MLLMs, including logical reasoning (LR), attribute reasoning (AR), relation reasoning (RR), fine-grained perception single instance (FP-S), fine-grained perception cross instance (FP-C), and coarse perception (CP). To validate that our model is an all-around player, the experiments on MMBench are conducted. The results, summarized in Table 6, show that our method achieves the highest overall performance, even outperforming methods specifically tuned for this benchmark, *e.g.*, mPlug-Owl, which leverages an advanced LLM LLaMA2 (Touvron et al., 2023b). Our model exhibits strong fine-grained perception ability. It can be attributed to the incorporation of the referential comprehension tasks during the multi-modal instruction tuning. The strong performance of our method on MMBench demonstrates its effectiveness in various aspects of multi-modal reasoning and comprehension.

### A.4 QUALITATIVE ANALYSIS

We show qualitative results on different types of vision-language tasks and RC tasks. The results of mPlug-Owl (Ye et al., 2023) and Shikra (Chen et al., 2023b) are from their official online demos.

**Knowledge-intensive QA.** Fig 4 depicts an example of this task, where the model is required to identify an album cover and provide an introduction for the album. mPlug-Owl can identify the album *The Wall* correctly but begins to hallucination when introducing this album. Some introductions provide by mPlug-Owl do not align with the actual facts associated with the album. The answer of Shikra focuses on the content of the image, indicating a poor instruction-following ability of Shikra. In contrast, our model not only correctly identifies the album but also delivers an accurate and relevant introduction, showcasing its strong instruction-following capability.

**Fine-grained QA.** As shown in Fig. 5, mPlug-Owl fails to output the correct number of people in the image. It may be because that its lack of RC ability hinders its fine-grained understanding of images. Despite Shikra having RC ability, it also fails to give a correct answer. This result can validate that trained with limited RC tasks Shikra struggles to achieve a fine-grained perception of images. In contrast, our model makes a correct answer to the question, demonstrating its strong capability in fine-grained image perception.

**Referential Comprehension Reasoning.** We show a case of RC reasoning in Fig. 6. In this case, our model accurately locates Steve Nash. Shikra seems to be unaware of who Steve Nash is. Surprisingly, Shikra's output coordinates point to the face of Dirk Nowitzki. We also need to point out the result of Shikra is generated using the instruction template that is identical to the one used during training. Shikra shows limited instruction-following ability when using a template that is different from the one used during training. Our model also shows a ability for multi-round conversation. It can understand what "him" refers to in the second question. As Shikra makes a wrong response for the first question, in the second question, we give the correct coordinates manually. However, Shikra

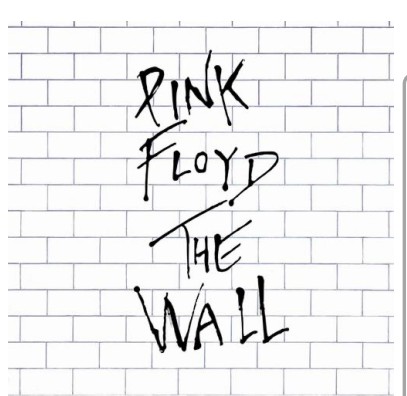
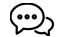

Tell me about the album.

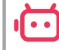

The image features a brick wall with the words "Pink Floyd" written on it, along with the phrase "Pink Floyd Wall." This could be a reference to the band's iconic album "The Wall," which was released in 1979. The album is known for its unique artwork and storyline, which features a character named Pink, who builds a wall to separate himself from the world. The album's cover art features a brick wall with the band's name written on it, and the brick wall in the image could be a nod to this iconic design.

(a) Our Method

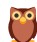

The album "The Wall" is a classic rock masterpiece by the British rock band Pink Floyd. Released in 1979, it is the band's ninth studio album and one of their most successful, selling over 23 million copies worldwide. The conceptual album is divided into 3 parts: "Part One: The Dark Side of the Moon," "Breathe," and "On the Run," each with its own theme and musical style.
The album explores the isolation and loneliness of its protagonist, Piggy, who is stuck behind "the wall," a symbol for the barriers that people create in their lives. It is considered one the most influential and iconic rock albums of all time, and its themes of alienation, mental health, politics, war, love, death, time and space continue to resonate with listeners decades after its release.

(b) mPlug-Owl

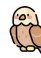

Pink Floyd wall, with white lettering on a brick wall, reads "Pink Floyd the Wall." The artwork is a black-and-white photo of a white brick wall, reminiscent of the iconic album cover of the same name. This design is likely a representation of the band's famous album, which was released in 1989 and became an enduring symbol of the music industry. The wall serves as a backdrop for the band's logo and artwork, creating a striking visual element that represents the iconic album.

(c) Shikra

Figure 4: A comparison of knowledge-intensive QA. We mark the hallucination part in red.

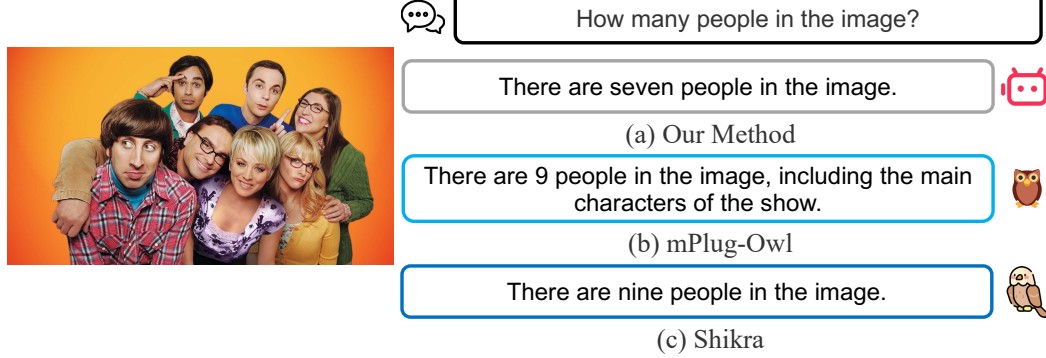

Figure 5: A comparison of fine-grained QA.

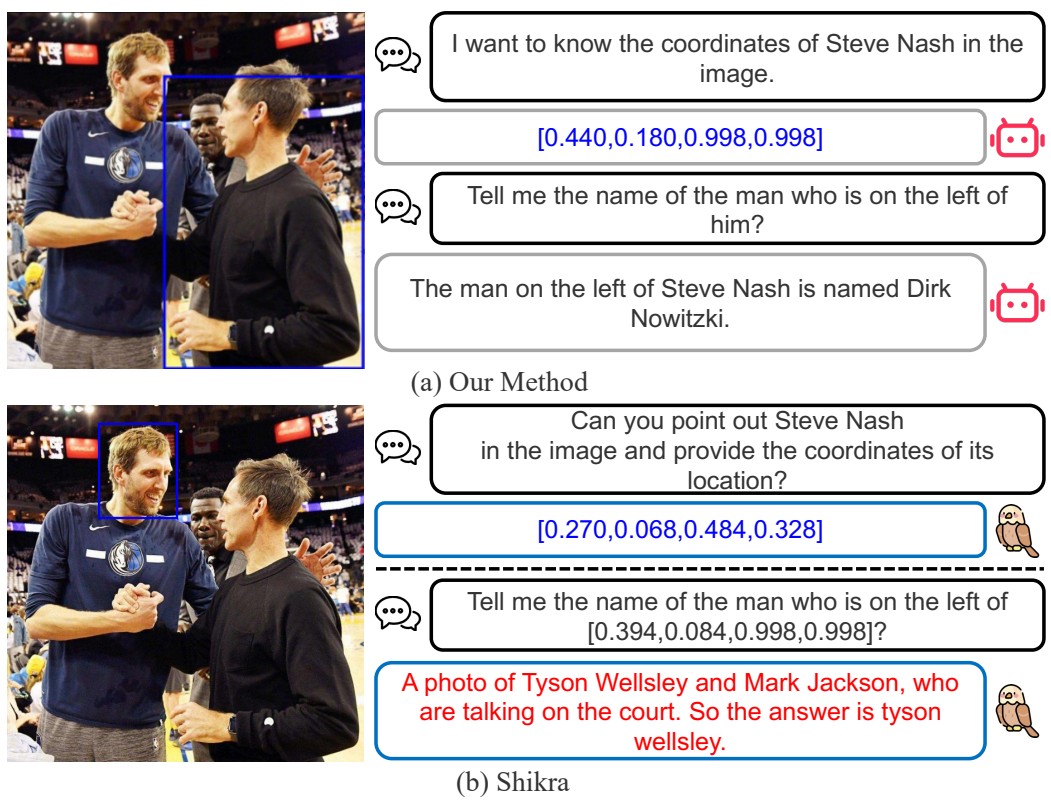

Figure 6: A comparison of referential comprehension reasoning. We mark the hallucination part in red.

fails to follow the instructions and begins to hallucination. These results can further validate the RC ability of our model.

**More Qualitative Results.** More qualitative results of our model are shown in Fig. 7. Our model demonstrates the ability to generate unique descriptions with contextual information when provided with coordinates of a specific area. For instance, instead of simply outputting "helmet", our model uses "woman's" features for differentiation. Moreover, our model successfully locates items mentioned in descriptions that require outside knowledge. For example, it can correctly identify what can be drunk in the image. In the last two cases, our model not only provides correct answers but also locates the mentioned items in the image. This ability can achieve more applications.

We also present qualitative results of our model on multi-round conversation about RC in Fig. 8 and Fig. 9. Our method can understand the complex referential relationships in the dialogue context, *e.g.*, it, her and this instrument. As shown in Fig. 9, although the model initially gives the wrong answer about what is holding by the man, when it is told that its answer is incorrect, the model can make corrections and provide the correct answer. This result can further demonstrate the promising instruction-following ability of our model. Additionally, these qualitative results highlight that the integration of RC ability significantly expands the range of tasks our model can successfully handle, thereby broadening its application potential.

**Qualitative Results of Generated Referring-expression-bounding-box Pairs.** We illustrate some examples of generated referring-expression-bounding-box pairs in Fig. 10. For each object in the image, the bounding box description bootstrapping method can generate a description related to that object. Most of generated descriptions are correct. However, some generated descriptions exhibit incorrect or ambiguous descriptions that fail to uniquely identify an object. As shown in Fig. 10 (a) and (b), when leveraging visual grounding to locate these descriptions in the image, IOU between the predicted bounding box and the ground-truth bounding box is low. Our self-consistent filtering method can effectively filter them out. These results can further validate the effectiveness of

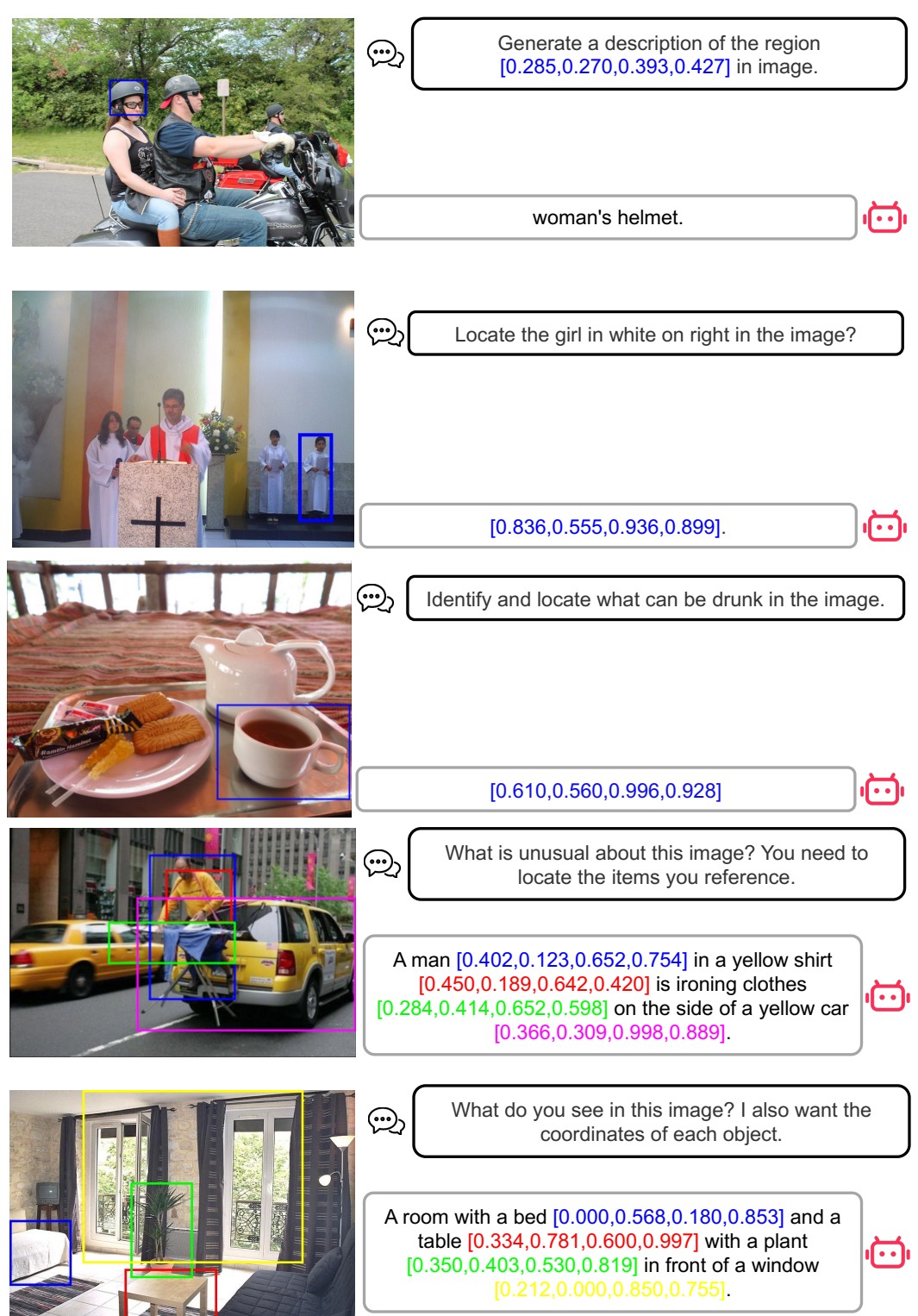

Figure 7: More qualitative results of our model.

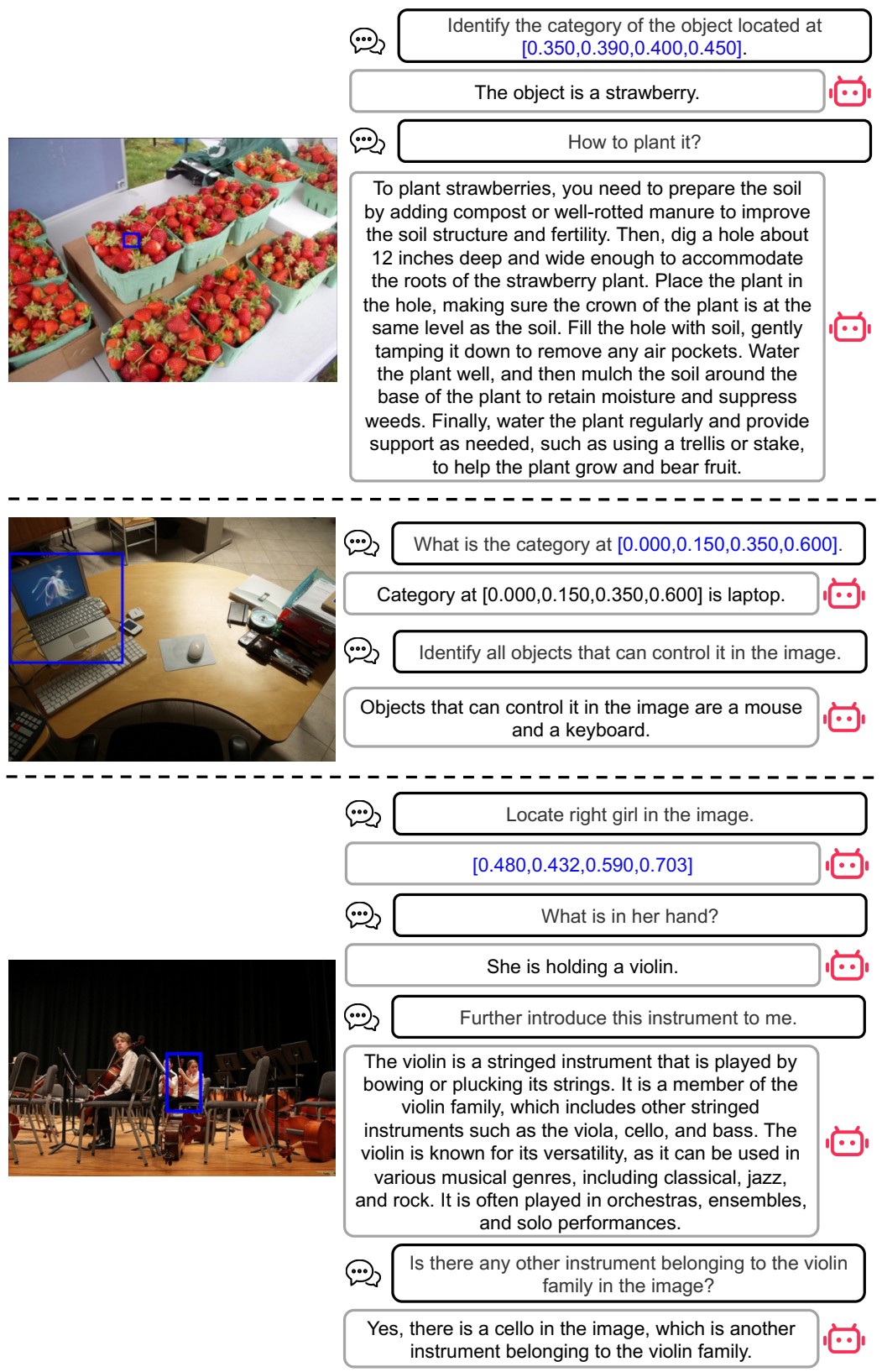

Figure 8: Qualitative results of our model on multi-round conversation.

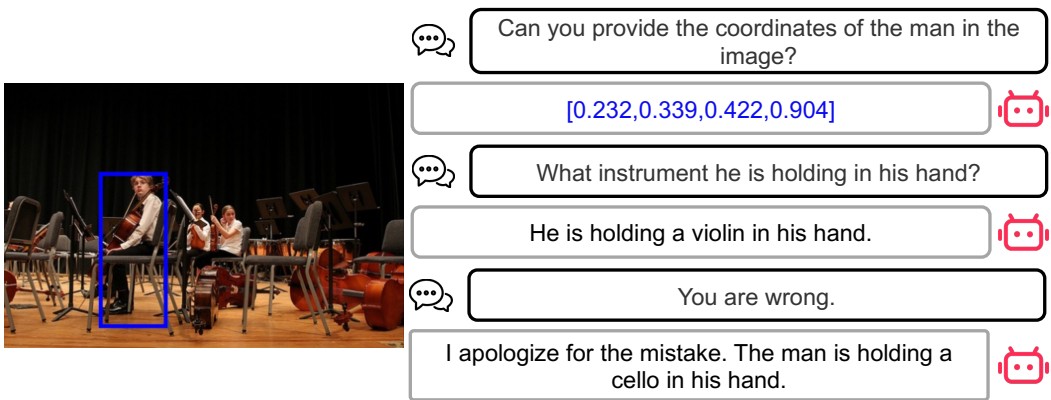

Figure 9: Qualitative results of our model on multi-round conversation with correction.

our self-consistent bootstrapping method to extend the object annotations to high-quality referring-expression-bounding-box pairs.

## A.5 INSTRUCTION TEMPLATES

**Instruction Templates of Visual Relation Reasoning.** We list all the instruction templates of visual relation reasoning task below. For the Task2, we use different instruction templates to ask the model output different contents (coordinates, class name, or both coordinates and class name).

---

**Instruction templates of visual relation reasoning-Task1**

What is the relation between <subject> and <object>?
Describe the relation between <subject> and <object>?
Assist me in finding the relation between <subject> and <object> in the photo.
In the given image, could you find and tell me the relation between <subject> and <object>?
I need help identifying the relation between <subject> and <object>. Can you point it out in this image?
What is the relation between <subject> and <object> in this picture?
Could you describe the relation between <subject> and <object> in this image?
I'm having trouble identifying the relation between <subject> and <object>/g. Could you clarify it for me in this image?
Can you help me understand the relationship between <subject> and <object> in this image?
I'm trying to understand the relation between <subject> and <object>. Can you help me by describing it?
I need some assistance in identifying the relation between <subject> and <object> in this image.
In this image, can you find and describe the relation between <subject> and <object> for me?
Could you please explain the relation between <subject> and <object>.

---

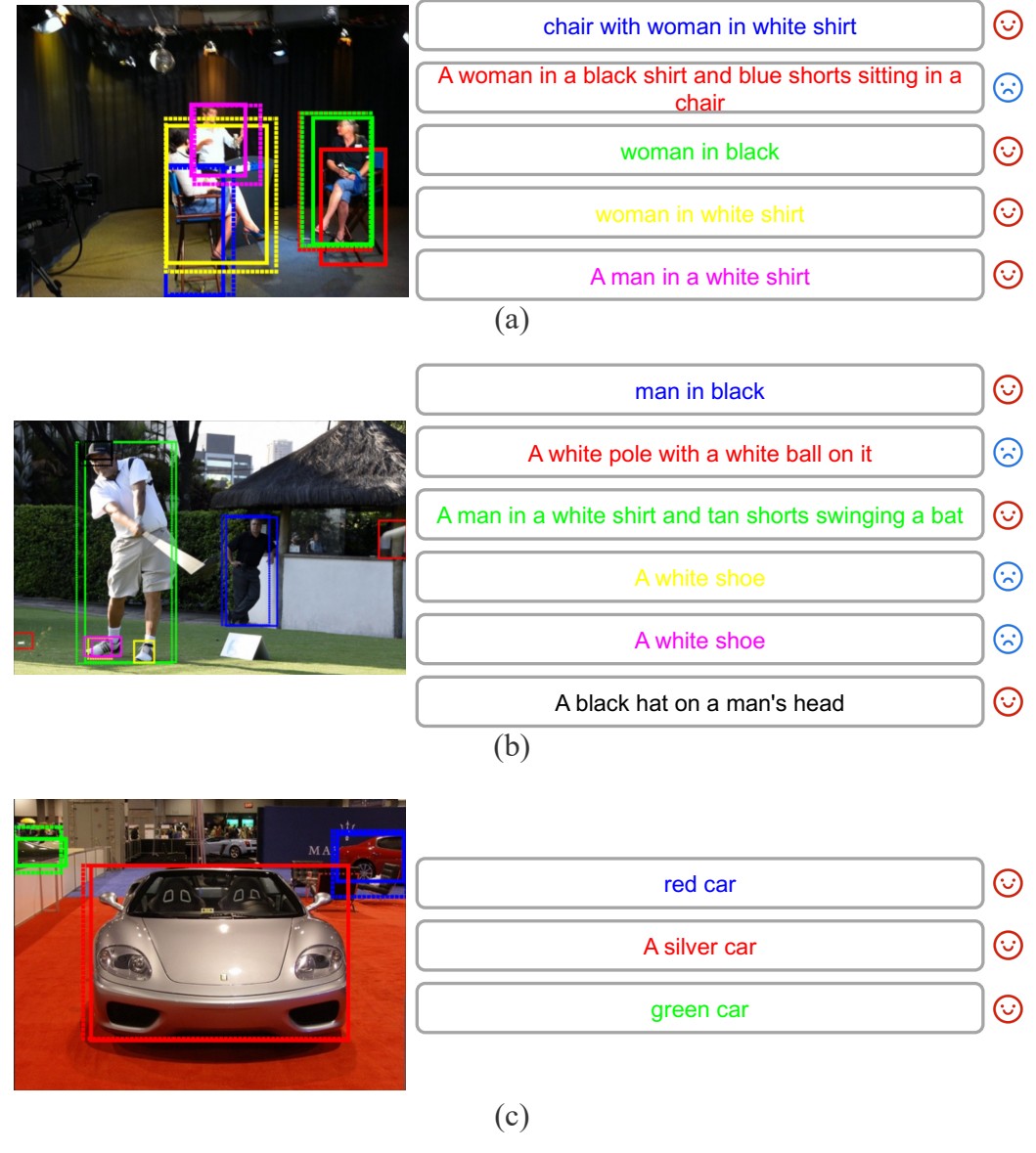

Figure 10: Qualitative results of generated referring-expression-bounding-box pairs. The solid rectangle represents the ground-truth bounding box. The dashed rectangle represents the bounding box generated by our model with visual grounding according to the generated description. ☹ denotes the generated description is filtered out by the proposed self-consistent method.

---

**Instruction templates of visual relation reasoning-Task2#1**

Assist me in locating the position of all the objects <relation> the <subject>?

I want to know the coordinates of all the objects <relation> the <subject>?

Detect all the objects have a relationship <relation> with the <subject> and output there locations.

There are some objects that are <relation> the <subject>. Could you tell me there locations?

Identify all the objects that have a relationship <relation> with the <subject>. Where are they located?

Please locate all the objects that are <relation> the <subject> and provide their coordinates.

Find all the objects that have a relation of <relation> with the <subject>. Can you give me their positions?

Point out the objects that are <relation> the <subject>. Where can I find them?

I need to locate all the objects that are <relation> the <subject>. Can you assist me with this task?

Could you help me find all the objects that have a relation of <relation> with the <subject>? Please provide their locations.

Please detect all the objects that are <relation> the <subject>. Output their positions.

Identify and provide the coordinates of all objects that are <relation> the <subject>.

Find the objects that have a relation of <relation> with the <subject>. Where are they situated?

What objects have the relation of <relation> with the <subject>? Could you locate them for me?

Can you help me locate all the objects that are <relation> the <subject> and give me their positions?

Output the positions of all objects that have a relation of <relation> with the <subject>.

Identify the objects that are <relation> the <subject>. Where are they located?

Please locate all the objects that are <relation> the <subject> and provide their positions.

---

**Instruction templates of visual relation reasoning-Task2#2**

Assist me in identifying the categories of all the objects <relation> by the <subject>?

Detect all the objects <relation> by the <subject> and output there categories, respectively.

There are some objects that are <relation> the <subject>. Could you tell me there categories?

I want to know the categories of all the objects whose relation is <relation> with the <subject>?

Identify the object categories that are <relation> the <subject>.

Find all objects that are related to <subject> using the relationship <relation>, and categorize them.

Your task is to recognize and classify all objects that are <relation> by the <subject>.

Please determine the categories of all objects that are <relation> by the <subject>.

Can you identify the categories of objects <relation> the <subject>?

Your job is to identify all objects that <relation> the <subject> and list their categories.

Detect and categorize all objects that are <relation> the <subject>.

I need you to determine the categories of all objects that <relation> the <subject>.

Identify and classify all objects that are <relation> the <subject>.

Please identify the categories of all objects that are <relation> the <subject>.

Please help me identify the object categories whose relationship is <relation> with <subject>.

Recognize and categorize all objects that are related to <subject> using the relationship <relation>.

I need you to categorize all objects that is related to <subject> with relationship as <relation>.

---

**Instruction templates of visual relation reasoning-Task2#3**

Your task is to locate all objects that have a relation <relation> with <subject> and classify them.

I need you to categorize and locate all objects that is related to <subject> with relationship as <relation>.

Please locate and categorize all the objects that have a relation of <relation> with <subject>.

Assist me in locating and classifying all the objects <relation> the <subject>?

Find all the objects that have a relation of <relation> with the <subject>. Can you give me their positions and categories

Your task is to locate all objects that have a relation <relation> with <subject> and classify them.

I need you to categorize and locate all objects that is related to <subject> with relationship as <relation>.

Please locate and categorize all the objects that have a relation of <relation> with <subject>.

Assist me in locating and classifying the position of all the objects <relation> the <subject>?

Find all the objects that have a relation of <relation> with the <subject>. Can you give me their positions and categories?

Your task is to locate and classify all objects that are related to <subject> using the relationship <relation>.

I need you to locate and categorize all objects having a relationship <relation> with the given <subject>.

Find all objects related to <subject> with the relationship <relation>. Categorize and locate them for me.

Your objective is to locate and classify the objects that are related to <subject> through the relationship <relation>.

I require you to detect and categorize all objects that have a relationship <relation> with <subject>.

Please find and classify all objects that has a relationship <relation> with <subject>.

Assist me in locating and categorizing all objects that related to <subject> with the relationship <relation>.

Find all objects that are related to <subject> using the relationship <relation>. Categorize and locate their positions.

Your task is to identify and classify all objects related to <subject> through the relationship <relation>.

I need you to locate and categorize all objects that have a relationship <relation> with <subject>.

Assist me in locating and classifying all objects that are related to <subject> through the relationship <relation>.

Find all the objects that has a relationship <relation> with <subject>. Categorize and locate their positions for me.

---

**Instruction Templates of Coarse Visual Spatial Reasoning.** We list all the instruction templates of coarse visual spatial reasoning task below. Similar to visual relation reasoning, different instruction templates are used to ask the model output different contents.

Instruction templates of coarse visual spatial reasoning#1

Identify the objects located at <loc> of <object>. Please classify them by category and provide their locations.
I need to know what objects are present at <loc> of <object>. Can you help me locate and categorize them?
Find all the objects at <loc> of <object>. Please provide me with their categories and locations.
I want to know the categories and positions of the objects located at <loc> of <object>.
Locate and classify all the objects at <loc> of <object>.
Could you tell me the categories and positions of the objects present at <loc> of <object>?
Help me locate and categorize all the objects at <loc> of <object>.
I need to know the categories and locations of the objects at <loc> of <object>.
What are the categories and positions of the objects located at <loc> of <object>?
Identify and locate all the objects at <loc> of <object>. I need their categories and positions.
I want to know the categories and positions of the objects at <loc> of <object>.
Locate and classify all the objects at <loc> of <object>. Please provide me with their categories and positions.

Instruction templates of coarse visual spatial reasoning#2

What are the categories of the objects located at <loc> of <object>?
Detect and classify all the objects at <loc> of <object>. I need to know their categories.
Please find and categorize all the objects present at <loc> of <object>.
Give the categories of all the objects you can find at <loc> of <object>.
I need you to find and categorize all the objects that are at <loc> of <object>.
Please provide me with the categories of all the objects present at <loc> of <object>.
What types of objects are located at <loc> of <object>? Please list their categories.
Please find all the objects at <loc> of <object> and give me their categories.
What are the categories of the objects that are present at <loc> of <object>?
I need you to classify all the objects located at <loc> of <object>.
Please give me the categories of all the objects that are located at <loc> of <object>.

Instruction templates of coarse visual spatial reasoning#3

What are the coordinates of the objects located at <loc> of <object>?
Detect and give the coordinates of all the objects at <loc> of <object>.
Please find and locate all the objects present at <loc> of <object>.
Give the detail locations of all the objects you can find at of <loc> <object>.
Locate all the objects and give there coordinates found at <loc> of <object>.
What are the positions of all the objects at <loc> of <object>?
Can you find and list the positions of all the objects present at <loc> of <object>?
Provide the coordinates of objects located at <loc> of <object>.
List and indicate the positions of all objects at <loc> of <object>.
Enumerate and specify the positions of all objects found at <loc> of <object>.
What objects are situated at <loc> of <object> and where precisely are they located?
What are the coordinates of all objects found at <loc> of <object>?

**Instruction Templates of Object Counting.** We list all the instruction templates of object counting task below. <category> will be replaced by the category name.

**Instruction templates of object counting#1**

Can you tell me how many <category> are present in this picture?
I need to know the number of <category> in this image.
Count how many <category> are in this picture.
Please determine the quantity of <category> shown in this image.
How many instances of <category> can you find in this picture?
I would like to know how many <category> are visible in this image.
Count the number of <category> that you see in this picture.
Please provide me with the count of <category> in this image.
How many objects of <category> are in this image?
Can you count the items of <category> in this picture?
What is the total number of <category> in this image?
How many <category> can you spot in this image?
Please determine the quantity of <category> in this image.
Count the number of <category> that appear in this picture.
How many <category> are in the picture?
Counting the number of <category> appeared in the image.
Please give me the number of <category> appeared in the image.

**Instruction templates of object counting#2**

How many objects in the image are of the same category as <object>?
Count the number of objects in the image that are similar to <object> in category.
What is the total count of objects that share the same category as <object> in the image?
How many objects in the image have the same category as <object>?
Count all the objects in the image that fall under the same category as <object>.
What is the number of objects that share the same category as <object> in the image?
Count the objects that belong to the same category as <object> in the image.
How many objects of the same category as the object represented by <object> appear in the image?
Count all the instances whose category is the same as <object> present in the image.

**Instruction Templates of Object Detection.** We list all the instruction templates of object detection task below. <category> will be replaced by the category name.

**Instruction templates of object detection#1**

Locate and mark the positions of all <category> in the image.
Find all the instances of <category> in the image and indicate their respective locations.
Spot and record the coordinates of every <category> present in the image.
Identify the <category> in the image and provide their precise locations.
Can you determine the positions of all the <category> in the image and list them?
Pinpoint the <category> in the image and give me their exact coordinates.
Locate all the <category> in the image and provide their locations in detail.
Detect and report the locations of all the <category> present in the image.
Find and list the locations of every <category> in the image.
Please identify the <category> in the image and give me their locations.
Provide me with the precise locations of all the <category> in the image.
Detect and record the positions of the <category> in the image.
Spot all the instances of <category> in the image and give me their coordinates.
Detect all the <category> in the image, and output there location.
There are some <category> in the image, could you help me to locate them and give me their coordinates.
What are the coordinates of the <category> in the image.
Give the detail locations of all the <category> you can find in the image.

Instruction templates of object detection#2

Locate all the items in the picture that share the same category as <object> and provide their coordinates.
Spot every object that belongs to the same category as <object> and indicate their positions.
Identify all the objects that fit the same category as <object> and display their coordinates.
Find all the objects that have a similar classification as <object> and output their locations.
Locate and report the coordinates of all the objects that share the category with <object>.
Detect all the objects in the image that have the same classification as <object> and provide their positions.
Spot all the objects that belong to the same category as <object> and show their coordinates.
Identify every instance that falls under the same category as <object> and report their locations.
Find and output the coordinates of all the objects that have the same category as <object>.
Locate all the objects in the picture that have a similar classification as <object> and display their positions.
Detect and report the positions of all the objects that share the category with <object>.
Spot every instance that has a similar classification as <object> and indicate its coordinates.
Identify all the objects that have the same classification as <object> and output their positions.
Find all the objects that belong to the same category as <object> and report their locations.
Locate and output the coordinates of all the items that have a similar category as <object>.
Detect all the instances in the image which have the same category with <object>, and output there location.
Detect and report the locations of all the instances present in the image, these instances should have similar category with <object>.
Given an <object>, please help me to find all the instances with the same category. The output should be the coordinates of detected instances.

**Instruction Templates of Multi-choices VQA.** Our instruction tuning dataset also includes the multi-choices VQA, *e.g.*, A-OKVQA. Therefore, we also construct some instruction templates for this task and list them below. Placeholder <options> will be replaced by the options.

**Instruction templates of multi-choices VQA**

Please take a look at the image and select the correct answer for <question> from the options given below \n<options>.

Examine the image and select the best matched answer to the question: <question> from the options given below\n<options>.

There are some options\n<options>. I have a question for you: <question> Can you select the best matched answers from the given options based on the image?

Regarding the image, you need to identify the correct answer to the question <question> from the given options\n<options>.

Analyzing the image, can you identify the best matched answer to <question> from the given options\n<options>.

Looking at the image, can you quickly answer my question: <question>. Some potential answers are given in the following options\n<options>.

Referring to the image, please select the answer for this question: <question> from the options\n<options>.

Could you please check the image and select the answer for my question: <question> from the options\n<options>.

Here is an image and a question: <question> for you. Please select an option that can answer the question from the given options\n<options>.

For this image, I want to know which option can answer my question: <question> correctly. The options are\n<options>.

Take a look at the image, can you select the best matched answer to the following question: <question> from following options\n<options>.

Considering these options\n<options>. I need a correct selection from these options that can answer this question: <question> in regards to the image.