# OpenReview forum: "Pink: Unveiling the Power of Referential Comprehension for Multi-modal LLMs"
_ICLR.cc/2024/Conference — ICLR 2024 Conference Withdrawn Submission_

### Official Review · Reviewer_GqeY · 2023-11-01

**Soundness:** 3 good
**Presentation:** 2 fair
**Contribution:** 2 fair
**Rating:** 3
**Confidence:** 5

**Summary:**

In this paper, the authors argue that existing Multimodal Large Language Models (MLLMs) lack the referential comprehension (RC) ability, i.e., identifying a specific object or area in images. Thus, they propose to represent the referring object with the coordinates of its bounding box and convert the coordinates into texts in a specific format. By regarding the coordinates as natural language, they can use a small instruction tuning dataset to transfer the knowledge in pretrained models. Furthermore, they propose a self-consistent bootstrapping method to extend dense object annotations into high-quality query-box pairs. The model is trained with a parameter-efficient tuning framework. Results show that the whole pipeline can significant improve the referential comprehension ability of MLLMs.

**Strengths:**

+ The problem of improving the referential comprehension ability of MLLMs is a very important topic. This work puts more emphasis on this worth exploring direction.

**Weaknesses:**

+ The novelty of the whole pipeline is limited. Although many techniques are used, all techniques are well-studied and straightforward. For example: 1) using parameter efficient training to avoid overfitting; 2) building instruction dataset for instruction tuning; 3) transforming bounding boxes into coordinates into the text. The only "new" thing may be the self-consistent bootstrapping method. From my understanding, it looks more like a trick to filter data. Overall, I think the whole contribution is very limited.

**Questions:**

Based on the results in Table 2, the Pink model without * (without generated query-box pairs) shows very limited performance gains. It would be better to have more explanations to demonstrate the effectiveness of the proposed architecture. Otherwise, it feels like that the main performance gains come from the generated new dataset (i.e., Object365 with generated pairs).

---

> ### Author Response · Authors · 2023-11-12
> **The clarification on the contributions and performance**
>
> 1) On the novelty and contributions of this work.
>
> Thank you for the constructive comments. We would like to clarify the main contribution of the paper. The paper did not claim that transforming bounding boxes into coordinates into the text or the use of parameter efficient training as the main contributions. We summarized the main contributions in the 5th paragraph in the introduction as “Our work is an original exploration to enhance LLMs with RC abilities by leveraging annotations from existing datasets and alleviating the dependence on expensive GPT-4 APIs.”.
>
> This paper argued that the visual encoder shall be tuned during multi-modal instruction tuning (using Adapter or LoRA) instead of being frozen. Then it boils down how to build a good instruction tuning set for RC tasks at a low cost. This paper proposed a new instruction set construction pipeline by designing diversified RC tasks using the annotations in existing datasets, without heavily using the API of GPT-4. Then plus self-consistent bootstrapping, the proposed method has achieved the best performance on both zero-shot and fine-tuning settings. Exploiting available annotations in existing datasets and using far less training data to boost the performance of RC capabilities of MLLMs are very useful techniques and the main contributions of the paper to share with the academic community.
>
> Moreover, Shikra [Chen et al. 2023b] was trained with limited RC tasks, which can only deal with the RC tasks included in the instruction-tuning. As shown in the qualitative results, our method shows fairly good generalization ability on various unseen RC tasks, e.g., the two questions in Fig.3. These results further validate the contribution and value of the proposed method.
>
> 2) On the performance gain.
>
> For the ablation studies, please refer to Table 1. It shows that without our designed RC tasks, the model shows limited instruction-following ability on the RC tasks. As more RC tasks are included, the model starts to exhibit better instruction-following ability for these tasks. The performance improves from 0.2\% to 63.1\% for PointQA. Moreover, the addition of the RC tasks also leads to 3.2\% performance gains on IconQA. As shown in Table 2, even without generated query-box pairs, our method outperforms the compared methods by a clear margin. For example, our method outperformed Shikra by about 6.0\% on OK-VQA.
>
> Given the intensive study on MLLMs, the authors do not think these performance improvements (+6.0\% on OK-VQA, +6.0\% on VSR and +4.7\% on IconQA as shown in Table 2) are limited. Every step forward over the SotA is very hard, especially given the data/computing resource required by the proposed method. In academia, we do not possess abundant data and GPUs, so IMHO the data and computation efficient method reaching SotA on some datasets is valuable to report to the community timely. Also the authors do not agree that these performance gains just come from the generated new dataset (i.e., Object365 with generated pairs).
>
> We sincerely appreciate your second thoughts on the technical contributions and performance gains of this paper. Thank you!

---

### Official Review · Reviewer_z9P4 · 2023-11-04

**Soundness:** 3 good
**Presentation:** 3 good
**Contribution:** 2 fair
**Rating:** 3
**Confidence:** 5

**Summary:**

This paper proposes a method to enhance the referential comprehension (RC) ability to identify a specific object or area in images for multimodal large language models. The proposed method constructs the instruction tuning dataset with various designed RC tasks at a low cost by unleashing the potential of annotations in existing datasets. The proposed method achieves good performances on the public datasets.

**Strengths:**

This paper proposes a method to treat the coordinates as natural language by representing the referring object in the image using the coordinates of its bounding box and converting the coordinates into texts in a specific format. The model is trained end-to-end with a parameter-efficient tuning framework that allows both modalities to benefit from multi-modal instruction tuning.

**Weaknesses:**

The novelty  is limited. The proposed methods convert the coordinates into texts in a specific format. This idea is widely adopted in the multimodal large language models for  specific objects, e.g.  [GPT4RoI: Instruction Tuning Large Language Model on Region-of-Interest], [VisionLLM: Large Language Model is also an Open-Ended Decoder for Vision-Centric Tasks].  The proposed methods just seem like the tricks of the object coordinates of the MLLMs

**Questions:**

1. Please highlight the contribution and novelty of the proposed methods.
2. Please add more details of the proposed adapters comparing the finetuning, LoRA and etc.

---

> ### Author Response · Authors · 2023-11-11
> **The contributions are on instruction tuning dataset construction and self-consistent dataset generation**
>
> With all due respect, this paper did not claim converting coordinates into specific texts as a contribution, nor the Adaptor method. As this paper studied how to enhance the referential comprehension (RC) ability for MLLMs, both coordinates to texts [Chen et al. 2023b] and the Adaptor [Housby et al. 2019] are the building blocks for our work, which were cited properly in the paper. We could use LoRA as an alternative adaptor.
>
> The main contributions of the paper are the proposed instruction tuning dataset construction and self-consistent dataset generation methods. As explained in the 3rd and 5th paragraph in the introduction, “Our work is an original exploration to enhance LLMs with RC abilities by leveraging annotations from existing datasets and alleviating the dependence on expensive GPT-4 APIs.”.
>
> These innovations aim to construct the instruction tuning dataset with a series of diversified RC tasks. Our method has achieved the best performance on both zero-shot and fine-tuning settings with a clear margin over previous SoTA, which could be readily integrated with other approaches. Given the intensive research on MLLMs, every step forward is hard and inspiring to the academic community, especially using \emph{less training data} and \emph{without} using much of GPT-4. We believe the proposed method is worthwhile to share with the community timely.

---

### Official Review · Reviewer_ArFY · 2023-11-06

**Soundness:** 3 good
**Presentation:** 3 good
**Contribution:** 2 fair
**Rating:** 5
**Confidence:** 3

**Summary:**

This paper presents Pink 1, a novel Multi-modal Large Language Model (MLLM) that enhances the Referential Comprehension (RC) capabilities of MLLMs.

Pink 1 leverages an existing method where the referring object in an image is represented using the coordinates of its bounding box, which are then converted into text in a specific format. This allows the MLLM to treat the coordinates as natural language.

The authors also propose a unique method to construct an instruction tuning dataset with a diverse range of RC tasks using annotations from existing datasets. This method allows for low-cost dataset construction.

Further, a self-consistent bootstrapping method is introduced to extend dense object annotations into high-quality referring-expression-bounding-box pairs. Pink 1 is trained end-to-end with a parameter-efficient tuning framework, resulting in fewer trainable parameters and less training data.

The experimental results demonstrate the superior performance of Pink 1 on both conventional vision-language tasks and RC tasks, highlighting the potential of this approach.

**Strengths:**

1. Solid experimental results that validate the proposed model's superior performance on both conventional vision-language tasks and RC tasks. The authors also provided a detailed comparison with other models under the fine-tuning setting, demonstrating the effectiveness of their model.

2. While the method of representing the referring object in an image using the coordinates of its bounding box is not new, the authors introduced innovative methods such as a unique instruction tuning dataset construction and a self-consistent bootstrapping method.

3. The paper is well-written and organized, providing clear definitions and explanations of the proposed model and methodologies.

**Weaknesses:**

1. As acknowledged by the authors, the Pink utilizes the approach of converting bounding box coordinates into text to understand the location of objects within an image. This technique, while not novel, could potentially impose limitations on the model's ability to perceive fine-grained details, particularly in complex images with numerous or overlapping objects.

2. The novel instruction tuning dataset construction method and the self-consistent bootstrapping method proposed in this study are innovative. However, their effectiveness is largely dependent on the quality and diversity of the existing datasets used. There might be limitations when dealing with less common or more complex RC tasks not covered in the existing datasets.

3. Although the model performs well on the tested datasets, it's unclear how well it would generalize to other types of RC tasks or datasets that are more complex or have different characteristics.

**Questions:**

N/A

---

> ### Author Response · Authors · 2023-11-12
> **Discussion on the generalization ability on different RC tasks.**
>
> Q1: The limitation on complex images with numerous or overlapping objects.
>
> R1: Yes, this remains a very challenging case to MLLMs. Most of MLLMs, if not all, including the proposed method, can not well handle “images with numerous or overlapping objects” yet.
>
> Q2: There might be limitations when dealing with less common or more complex RC tasks not covered in the existing datasets.
>
> R2: As shown in Table 2, our method has achieved the best performance on the RC tasks under both zero-shot and fine-tuning settings. Any progress on the SotA for MLLMs is hard given current intensive efforts on MLLMs, especially our method requires less training data and computation resources. These results can validate the effectiveness of the proposed instruction tuning dataset construction method.
>
> Our method exhibits a stronger generalization ability on various RC tasks. The compared model Shikra can only deal with the RC tasks included in its instruction tuning stage. As shown in the qualitative results, our method shows fairly good generalization ability on various unseen RC tasks, e.g., the two questions in Fig.3. Certainly, there are more complex RC tasks to figure out in the future research, as we do not claim the proposed method can handle all RC tasks not covered in the existing datasets.
>
> Q3: It's unclear how well it would generalize to other types of RC tasks.
>
> R3: Current datasets and tasks for multimodal RC tasks are limited. It is hard, if not impossible, to provide a quantitative analysis for other types of RC tasks due to the lack of a benchmark.
>
> For the instruction tuning, in order to increase the diversity of RC tasks, we propose an efficient instruction tuning dataset construction method. Moreover, we also showed many qualitative results about the RC tasks in the main paper and Appendix.